# Cloning of the *DlERF10* gene from *Diospyros lotus* L. and cold tolerance analysis of the *DlERF10* gene in transgenic tobacco plants

**Ruijin Zhou**[1,2*], **Shuda Li**[1], **Xiaona Zhang**[1], **Yingying Wang**[1], **Huiling Hu**[1]

**1** School of Horticulture Landscape Architecture, Henan Institute of Science and Technology, Xinxiang, Henan, China, **2** Henan Province Engineering Research Centers of Horticultural Plant Resource Utilization and Germplasm Enhancement, Xinxiang, Henan, China

* persimmonzhou@163.com

## Abstract

In the north of China, *Diospyros* plants are vulnerable to low-temperature damage in winter and is considered as a major factor restricting the development of the persimmon industry in Northern China. *Diospyros lotus* L. is featured by high survival potential of seedlings, cold tolerance, and grafting affinity with D. *kaki* Thunb. *D. lotus* has been frequently used as rootstocks for *Diospyros spp*. ERF transcriptional factors are a subfamily of the AP2/ERF gene family and play an important role in plant growth and stress tolerance. To explore the structure and function of the ERF transcription factors in *D. lotus*, we performed RT-PCR to clone *DlERF10* from the leaves. The *DlERF10* gene was 1104 bp long, encoding 367 amino acids. In order to deeply study the cold tolerance of *DlERF10* gene, the *pBI121-DlERF10* overexpression vector was constructed, and agrobacterium-mediated transformation was carried out to transfer the gene into tobacco plants. The wild-type and transgenic tobacco plants were subjected to low-temperature stress. The results showed that the transgenic plants were less severely damaged by low-temperature stress than the wild-type plants. Besides, the SOD, POD and CAT activities of leaves enhanced, and PRO contents of leaves increased, while the MDA content decreased. It was concluded that the *DlERF10* gene increased the activity of protective enzymes in tobacco plants, thereby strengthening the tolerance to low-temperature stress. The present study proposes a candidate gene for engineering cold stress tolerance in *Diospyros* spp.

## Introduction

Plants are continuously challenged by several environmental stresses that impair their growth and production performances [1]. Abiotic stresses such as drought, cold, heat, and high salt greatly affect plant growth and development, yield and quality, and even limit geographical distribution [2]. Transcription factors (TFs) are a group of proteins specifically binding to the *cis*-acting elements in the promoter region of eukaryotic genes and regulating the expression intensity of target genes under specific circumstances. TFs are critical to the growth and environmental response of plants [3], regulating plant development and their responses to adverse environmental stresses [4–6]. Such as WRKY TFs can directly bind to the W-box

**Data availability statement:** All relevant data are within the manuscript and its Supporting Information files.

**Funding:** The author(s) received no specific funding for this work.

**Competing interests:** The authors have declared that no competing interests exist.

sequence ((T)(T) TGAC (C/T)) in the promoter region of the downstream target gene [7], and activate or inhibit the transcription of the target genes by interacting with the target protein [8]. They may up-regulate the expression of stress-related genes through integrating signal pathways mediated by abscisic acid (ABA), ethylene(ET), salicylate (SA), jasmonic acid (JA) and reactive oxygen species (ROS), thus playing a vital role in regulating plant response to abiotic stresses [9,10]. The study showed that WRKY TFs were widely related to plant tolerances to salinity [11–13], heat [14,15], cold [16] and drought stress [17]. MYB TFs contain a DNA-binding domain (DBD), a transcription regulation domain (TRD), an oligomerization site (OS), and a nuclear localization signal [18]. MYB TFs can bind to the downstream target gene promoter *cis-acting* elements MYBCORE and AC-box alone or through interaction with other proteins after being activated by environmental signals, and participate in regulating the expressions of downstream target genes, thereby regulating plant tolerances to stresses [19,20]. NAC is consisting of a highly conserved N-terminus and a relatively variable C-terminus. NACs regulate the flower organ development [21], flowering time [22], leaf senescence [23], fruit ripening [24], secondary cell wall formation [25], root development [26], and are also involved in plant responses to various biotic and abiotic stresses [27–29]. bHLH family of transcription factors is the second largest family of transcription factors in plants after the MYB. bHLH TFs are functionally diverse and can regulate plant tolerance to abiotic stresses such as low temperature and drought and biotic stresses such as pests and diseases [30,31]. The most studied bHLH TFs in cold stress response is ICE [Inducer of CBF (C‑repeat binding factor) expression], which specifically binds to the MYC cis-acting element in the promoter region of the CBF/DREB1 (Dehydration‑responsive element binding 1) gene. It can specifically bind to the MYC cis-acting element of CBF/DREB1 (Dehydration‑responsive element binding 1) to activate the CBF gene, and CBF specifically binds to the CRT/DRE element in the promoter region of the cold-regulated gene COR (Cold‑regulated) to activate the expression of the COR gene and improve the cold tolerance of plants [32]. bZIP has a conserved domain consisting of about 60–80 amino acids, including a highly conserved alkaline region and a relatively variable leucine zipper region [33]. When plants are stimulated by stress signals, bZIP is phosphorylated by the upstream signal-responsive kinases [34], and its stability is enhanced through phosphorylation [35]. Under stresses such as drought, salt, high and low temperatures, bZIP binds to the promoter regions of stress-related genes and interacts with other proteins to promote or inhibit the expressions of these genes, thus positively or negatively regulating the responses to abiotic stress and biotic stress [36–39]. bZIP not only regulates plant stress response, but also in roots department development, leaf formation, flower development and seed germination plays an important role [40].The AP2/ERF(APETALA2 and ethylene responsive element binding proteins) transcription factor superfamily is among the transcription factor families that have the largest number of members in plants [41]. An AP2/ERF transcription factor contains at least one distinct AP2/ERF structural domain, which is composed of 50–60 amino acids and where the amino acid residue region is highly conservative [42]. The GBD (GCC-box binding domain) structure of the protein is composed of three antiparallel β-sheets and an α-helix. The GBD structure binds specifically to the ethylene-responsive heterologous promoter GCC-box [43]. Depending on the number or structure of the conserved domain in the AP2/ERF transcription factor, the AP2/ERF family is divided into five subfamilies, namely, AP2, ERF, DREB, RVA, and Soloist [44]. Joufku [45] was the first to isolate the APETALA2 (AP2) transcription factor from *Arabidopsis thaliana* (L.) Heynh. (Fam.: Brssicaceae). Since then, AP2/ERF transcription factors have been isolated and identified from 20 plant species. The largest number of AP2/ERF transcription factors were identified in oilseed rape [46], totaling 531, followed by tobacco (375) [47], maize (214) [48], rice (170) [49], *A. thaliana* (147) [50], tomato (134) [51], and pineapple (97) [52] successively.

The EFR subfamily has the largest number of members among all subfamilies of the AP2/ERF superfamily [53]. The EFR subfamily transcription factors usually act as signal elements at the terminal of the ethylene signaling pathway. They regulate the expressions of target genes by binding to the *cis-acting* elements in the promoter regions, thereby participating in the stress responses of plants [54]. In *A. thaliana*, the overexpression of the ERF family gene can enhance the tolerance of transgenic plants to different abiotic stresses to a greater or lesser extent [55,56]. Some AP2/ERF transcription factors can regulate the expressions of key enzymes in proline biosynthesis, thereby strengthening plants' cold tolerance [57]. In one study, the *TaERF1* gene was downregulated in wheat due to drought and was transferred to *A. thaliana*, resulting in higher drought tolerance and also higher tolerance to high salt and low temperature [58]. An overexpression of OsERF115/AP2EREBP110 was conducive to enhancing rice's water retention and leaf-cooling capacity, which further improved overall tolerance and drought tolerance [59]. Melatonin can induce upregulation of the transcription factor OsERF53 in rice seedlings under salt stress, mitigating the inhibitory effect of salt on rice growth and development [60]. The *ZmERF1* gene in maize is involved in regulating hypoxia tolerance under waterlogging stress [61]. Salt stress induces a significant upregulation of the *ZmERF1* gene and has a positive regulatory effect on salt tolerance in maize [62]. Flesh lignification is a unique response to low temperatures in plants that causes deterioration in the quality of stored red-fleshed loquat fruits. *EjERF39* and *EjMYB8* in loquat fruits form a co-activator complex, which is capable of transactivating the promoter in the lignin biosynthesis gene *Ej4CL1*. These two transcription factors have shown similar expression patterns of lignification-related genes in the red-fleshed loquat variety 'Luoyangqing' and the white-fleshed loquat variety 'Ninghaibai' during postharvest treatment, which are considered to undergo different lignification processes. Another study has shown that *EjERF39* interacts with *EjMYB8* to regulate the lignin biosynthesis gene *Ej4CL1*, thereby enhancing the low-temperature tolerance of loquat fruits [63]. Zhang [64] found that the transcription factor *TERF2/LeERF2* in tomatoes was upregulated under low-temperature stress. After being transferred to tobacco, this transcription factor regulated the expressions of cold tolerance -related genes (e.g., *NtERD10B* and *NtERD10C*), which led to higher low-temperature tolerance of tobacco. Sun [65] cloned the *VaERF080* and *VaERF087* genes in grapes and transferred them into *A. thaliana*. They found that these two transcription factors improved the cold tolerance of *A. thaliana* by increasing the activity of antioxidant enzymes and upregulating the expressions of cold tolerance-related genes. Dai [66] showed that overexpression of the *VvERF2* gene promoted callus growth of grapes and the accumulation of secondary metabolites, such as phenols, which further promoted salt tolerance of grapes. *PtrERF108* from trifoliate orange regulates raffinose synthesis by regulating the raffinose synthase-encoding gene *PtrRafS*, thereby enhancing cold tolerance. *PtrERF108* overexpression led to higher cold tolerance of transgenic lemon plants, and the virus-mediated gene silencing of *PtrERF108* dramatically enhanced cold intolerance in trifoliate orange [67]. *BpERF13* in birch strengthened cold tolerance by upregulating the *CBF* gene and reducing reactive oxygen species [68]. From some cold-tolerant plants, such as *Tetrastigma hemsleyanum* Diels & Gilg [28], *Panax ginseng* C.A. Mey [69], and *Juglans mandshurica* Maxim. [70], researchers have identified several candidate cold-stress-responsive genes belonging to the AP2/ERF family.

*D. kaki*, a native plant species of China, has a cultivation history of more than 2000 years. China is rich in *Diospyros* resources, and 57 *Diospyros* species and 6 variants can be found in China [71], typically in tropical and subtropical regions. But in the north of China, *Diospyros* plants are vulnerable to low-temperature damage in winter. Low temperature is considered a major factor restricting the development of the persimmon industry in Northern China. *D. lotus* (https://www.plantplus.cn/info/Diospyros%20lotus?t=foc) is a *Diospyros* species

belonging to the Ebenaceae family [72] and is known for its high seedling survival rate [73], cold tolerance [74], and high grafting affinity with *D. kaki* [75]. Because of these advantages, *D. lotus* has been widely used as rootstock for *Diospyros* species. ERF transcription factors are involved in regulating plant growth and stress tolerance, though their roles in *Diospyros* species remain largely unknown. In this study, the *DlERF10* gene was identified from the young leaves of *D. lotus* and cloned. The cloned sequence was analyzed using bioinformatics, and the *DlERF10* overexpression vector was constructed and transferred into tobacco plants. The phenotype and the physiological and biochemical properties of the transgenic plants were analyzed to determine the roles of *DlERF10* in cold tolerance. Our results provide valuable clues for understanding the gene regulatory network and molecular mechanism of *DlERF10* in the cold tolerance of *D. lotus*.

## Materials and methods

### Materials and processing

The seeds of *D. lotus* were collected from Zhangcun Town, Huixian City, Henan Province (altitude 336.2 m, E113°52', N35°33'). After collection the seeds were soaked in alcohol for 30 s, and then disinfected with 9% sodium hypochlorite for 15 min. The seeds were sown in a nutrient bowl. When 3–4 new leaves grew out, seedlings with consistent growth status were chosen for carrying out different stress treatment experiments. The stress treatments were: low-temperature at 4°C, drought using 20% PEG6000, and salt using 250 mmol/L NaCl. Sampling for low-temperature stress treatment were done at intervals of 0 h, 6 h, 1 d, 3 d, and 5 d. However, sampling for drought and salt stress treatments were carried out at 0 h, 6 h, 12 h, 1 d, 3 d, 5 d, and 7 d intervals. Three seedlings were chosen for each time point in each stress treatment. Leaves were picked from *D. lotus* seedlings at 2nd–3rd leaves from the bottom to top, immediately placed in liquid nitrogen, and stored at −80°C.

### Cloning and bioinformatic analysis of the *DlERF10* gene

RNA extraction from young leaves was performed using the Trizol reagent. The first strand of cDNA was synthesized by RNA reverse transcription using the RevertAidTM reverse transcription kit. Based on the identified *ERF* gene sequences, specific primers DlERF10-F and DlERF10-R were designed (Table 1). PCR amplification was performed using cDNA as a template. The 20 μL PCR reaction system consisted of the following: cDNA template 1 μL (<200 ng), 2 × Taq PCR Master Mix II 10 μL, primers DlERF10-F and DlERF10-R 0.5 μL each,

**Table 1. Primer sequences.**

| Primer name | Primer sequence (5' to 3') |
| --- | --- |
| DlERF10-F | AGAAGAAGCAAGTATGTGTGGC |
| DlERF10-R | TCCGTCCTGCCATCTCCTAGAA |
| qDlERF10-F | ATCTGGAAGGGGGACAATTC |
| qDlERF10-R | AGAGTAGCGCGGCAAAATTA |
| DkActin-F | GCCATCATTAATTGGAATGGAAGC |
| DkActin-R | GTGCCACAACCTTGATCTTCA |
| NPTll-F | GATGGATTGCACGCAGGTTC |
| NPTll-R | ATATCACGGGTAGCCAACGC |
| F35s | GGAAGGTGGCTCCTACAAATGC |
| NtActin-F | CCTGAGGTCCTTTTCCAACCA |
| NtActin-R | GGATTCCGGCAGCTTCCATT |

supplemented with ddH$_2$O until 20 μL. PCR procedure followed: Predenaturation at 95°C for 2 min; denaturation at 95°C for 30 s, annealing at 56°C for 30 s, extension at 72°C for 1 min, 35 cycles; final extension at 72°C for 5 min. After the reaction, the PCR-amplified products were recovered and ligated to the pMD18-T vector to construct the recombinant plasmid. Next, *E. coli* DH5α competent cells were transformed with the recombinant plasmid, and the positive clones were screened and sequenced.

The cDNA sequences thus obtained were aligned against the homologous sequences using the NCBI BLAST program. The ORF and amino acid sequence of the gene were analyzed using DNAMAN 6. 0. 3. The phylogenetic tree was constructed using the MEGA 5.0 software. The conserved domain structure of the protein was predicted using the NCBI conserved domain search and Pfam 31. 0. Subcellular localization was performed using online software, including CELLO v. 2.5, PSORT Prediction, and SoftBerry ProtComp 9. 0 [55,76,77]. The 2,000-bp sequence upstream of *DlERF10* was regarded as the promoter region and extracted from the *D. lotus* genome database (http://persimmon.kazusa.or.jp/blast.html) by TBtools and submitted to the Plant CARE database for identifying the *cis-acting* elements.

### Specificity of *DlERF10* gene expression under environmental stresses

RNA was extracted from the samples in each of the above stress treatments in section 1.1, using the versatile plant RNA extraction kit (DNase I, CW2598S). cDNA synthesis was performed using the RevertAidTM 1$^{st}$ cDNA Synthesis Kit (#K1622, Thermo Scientific), and the product was stored at −20°C. *DkActin* was used as the internal reference gene. The *DlERF10* gene expressions under different abiotic stresses were determined using SYBR *Premix Ex Taq* II (Tli RNaseH Plus). The 10 μL qRT-PCR consisted of the following: SYBR premix Ex TaqTM II 5 μL, Primer-F 0.4 μL, Primer-R 0.4 μL, cDNA 0.7 μL, and ddH$_2$O 3.5 μL. The reaction procedure consisted of the following steps: 95°C 30 s, 95°C 5 s, 56°C 30 s, 40 cycles, 95°C 15 s, 60°C 30 s, 95°C 15 s. Four replicates were set up for each sample. The expressions of the target gene in different stress treatments were calculated using the $2^{-\Delta\Delta CT}$ method [78].

### Expression vector construction and transfer into tobacco

The plasmid vector was constructed using one-step cloning. The *E. coli* DH5α competent cells were transformed with the *pBI121-DlERF10* overexpression vector. Based on PCR validation, positive clones were picked and submitted to sequencing by Shanghai Shenggong Biological Company. The GV3101 Agrobacterium electro-competent cells were transformed with the recombinant pBI121 plasmid with the correct sequence. *Nicotiana benthamiana* Domin was transformed using the leaf disc method. The plants carrying the tolerance gene were screened using the kanamycin-containing culture medium. PCR validation was performed using primers of the target gene, universal primer 35s-F, reverse primer of the target gene, and NPT II primers. Positive plants were identified based on the results for the three pairs of primers. T2 transgenic tobacco plant was obtained by self-cross and screening and used for further experiments.

### Cold tolerance analysis of transgenic tobacco plants

Seeds of wild-type and transgenic tobacco plants were harvested and disinfected with 75% alcohol for 1 min and 9% sodium hypochlorite for 10 min. The seeds were washed with sterile water, placed on the 1/2MS solid medium, and incubated in an illumination incubator at 22°C, with a relative humidity of 50–60% and light intensity of 10,000 Lx, under a 16/8 h light/ dark cycle. After incubation for 7 d, seedlings with consistent growth status were selected and transferred to a nutrient bowl. Seedlings after 30 days of growth were subjected to

low-temperature stress at 4°C. At 0 h, 12 h, 1 d, 3 d, 5 d, and 7 d of the treatment, leaves were harvested from the same position in different plants and immediately placed in liquid nitrogen at −80°C. Three biological replicates were set up. The contents of MDA and PRO and the activities of SOD, POD, and CAT in leaves were determined using the kits manufactured by Ruixin Tech.

## Data processing

Analysis of variance was conducted using SPSS 25.0. Means were compared across the plants using Duncan's test. Data were analyzed and plotted using Excel.

## Results

### *DlERF10* gene clones and sequence analysis

One target sequence was cloned from the leaves of *D. lotus* using RT-PCR. The product was detected by 1.0% agarose gel electrophoresis. The target sequence was about 1100 bp in length (Fig 1). The target sequence was recovered and aligned with the *ERF* gene sequence in the gene pool. This newly identified gene was named *DlERF10*. The conserved structural domain of the *DlERF* gene was analyzed. The results showed that *DlERF* contained one conserved AP2 domain lying between the 95th to the 158th nucleotides. Therefore, *DlERF* belonged to the AP2 family.

### Bioinformatic analysis of the *DlERF10* gene

*DlERF10* gene contains an ORF of 1104 bp in length encoding 367 amino acids, with a protein molecular weight of 41.48 kDa and a theoretical isoelectric point of 9.37. It is a hydrophilic protein without a signal peptide structural domain, with a subcellular localization predicted

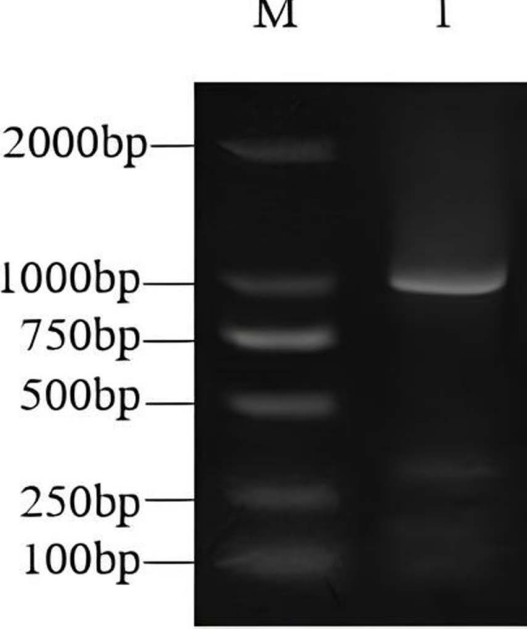

**Fig 1. Result of PCR product of the purpose gene.** M: DNA marker DL2000; 1: *DlERF10* gene clone fragment of *D. lotus*.

to be in the nucleus, and a secondary and tertiary structure consisting mainly of an irregularly convoluted and α-helix. The amino acid sequence of the DlERF10 protein was compared with the amino acid sequences of nine ERF proteins, including *D. kaki* (AGC79344.1), *D. oleifera* Cheng (EVM0025771.1), *Parasponia andersonii* (Planch.) Planch. (PON40526.1), *Tripterygium wilfordii* Hook. f. (XP_038684704.1), *Actinidia chinensis* Planch. (ADJ67433.1), *Pistacia vera* L. (XP_031287991.1), *Gossypium hirsutum* (XP_016702145.1), *Prunus persica* (XP_007200418.1), and *Malus domestica* Borkh. (NP_001306945.1) (Fig 2). The results showed that *D. lotus* had the highest amino acid homology with *D. kaki* belonging to the same genus of the same family, which was 83.85%. Besides, all of the 20 protein sequences contained the conserved AP2 domains (the part underlined in black in Fig 3). The 14th amino acid of this structural domain was alanine (A14). and the 19th amino acid was aspartic acid (D19), which are typical of the ERF family proteins [79]. It was presumed that the DlERF10 protein belonged to the ERF subfamily of the AP2/ERF family. Therefore, the protein was named *DlERF10*.

To further understand the genetic relationship between the species, we downloaded the amino acid sequences of 20 plant proteins homologous to the DlERF10 protein from the

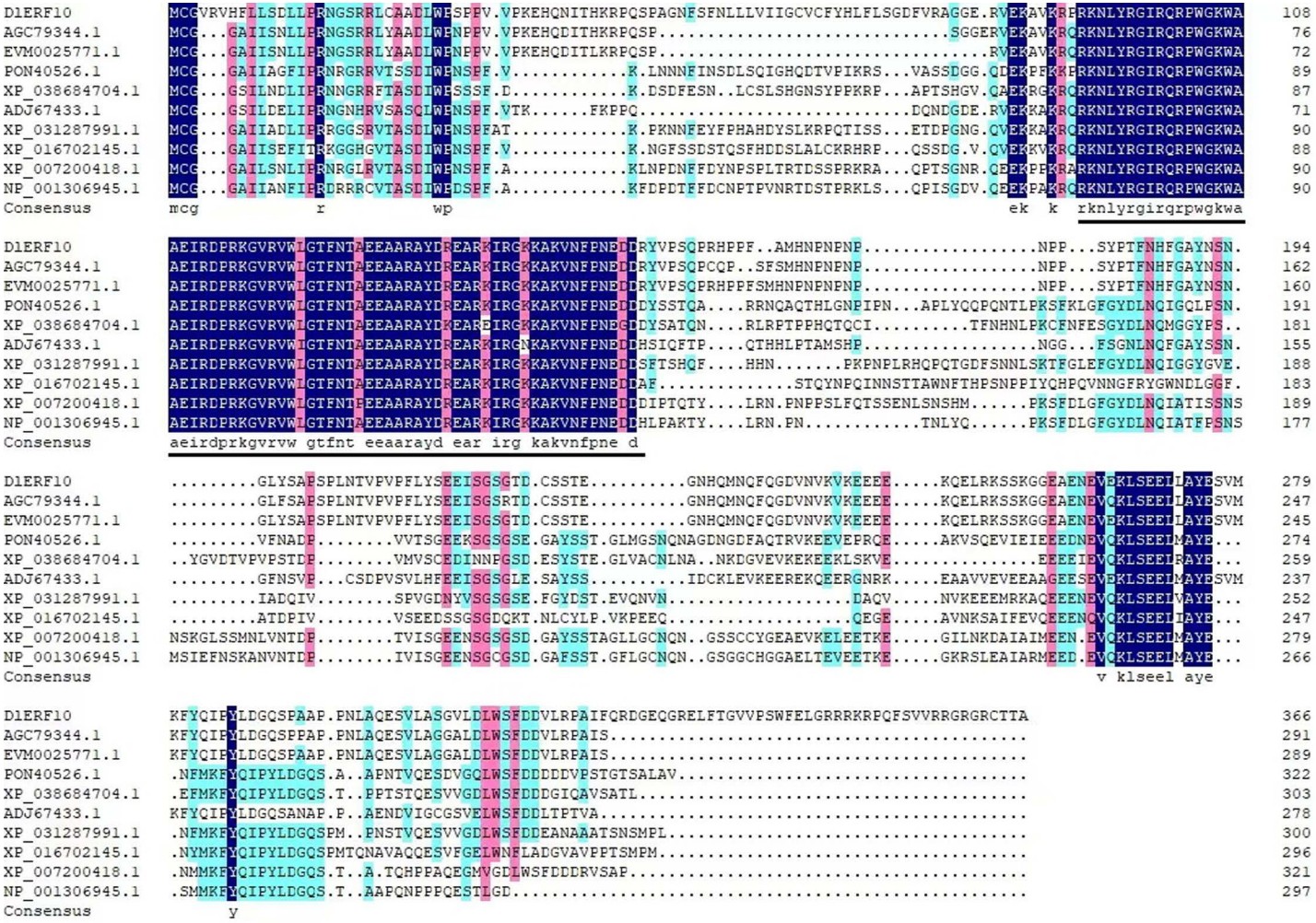

**Fig 2. Comparison of multiple sequences between DlERF10 and other species ERF protein.** The black line represents the conserved AP2 domain.

NCBI database. A phylogenetic tree was constructed using the neighbor-joining method (NJ) in the MEGA 5.0 software (Fig 3). The results showed that *D. lotus* had the closest genetic relationship with *D. oleifera* and *D. kaki* belonging to the same genus of the same family in terms of the DlERF10 protein; *D. lotus* had the most distant genetic relationship with *Populus alb*a L., *Pistacia vera*, and *Parasponia andersonii*.

The 2,000 bp nucleotide sequence upstream of the *DlERF10* gene was analyzed by Plant CARE to obtain the regulatory elements (Table 2). The prediction results showed that this sequence contained not only a large number of core sequence (TATA-box) in the promoter, the CAAT-box in the promoter and enhancer regions, and other basic promoter elements across eukaryotes, but also included 14 light-responsive elements (G-box, Box 4, I-box, GATA-motif, TCT-motif and AE-box), 8 hormone response-related cis-elements (TGA-box, P-box, TCA-element, ABRE, TGACG-motif and CGTCA-motif), and 2 growth and development related elements (CAT-box and circadian).

## Expression analysis of the *DlERF10* gene under abiotic stresses

qRT-PCR was performed to detect *DlERF10* expressions under abiotic stresses. The results showed that under all three types of stresses, *DlERF10* was upregulated than before. Under low-temperature stress, the *DlERF10* expression in leaves increased most significantly at 6 h, reaching the peak at this time point. The expression level was 6.39 times the expression in the

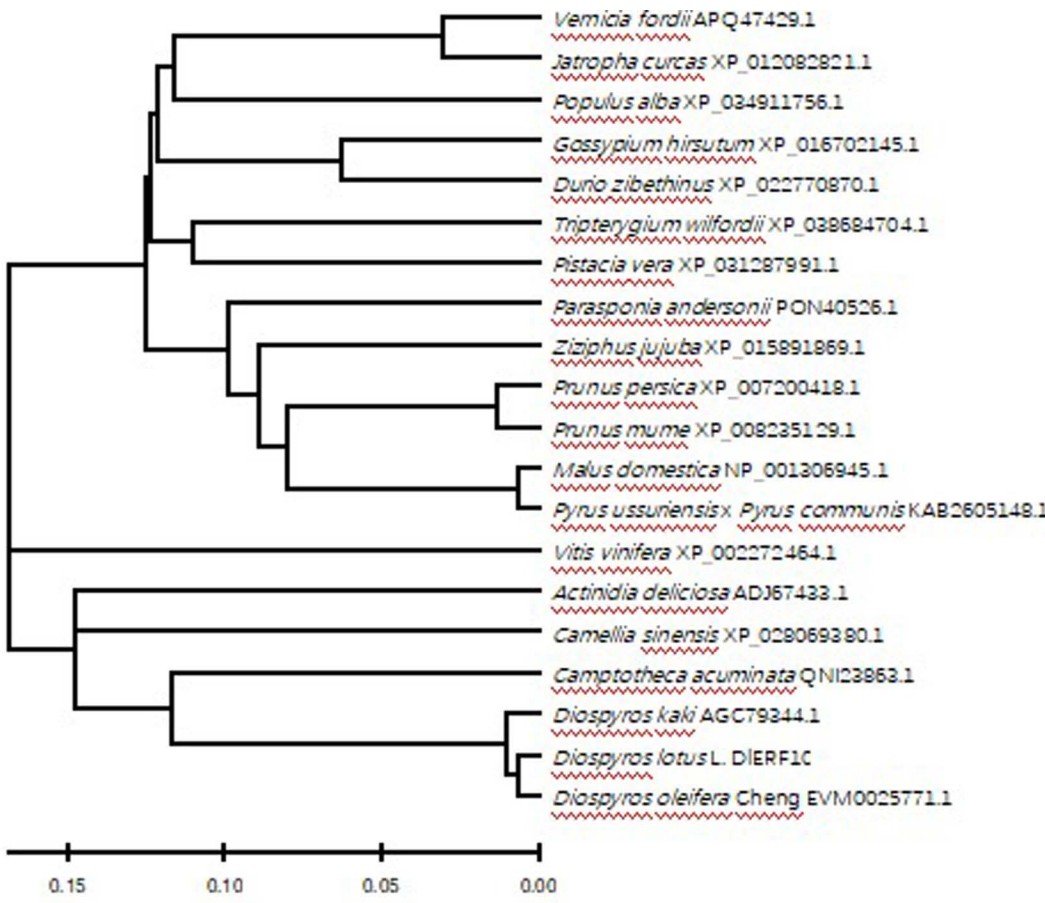

**Fig 3. Phylogenetic tree analysis of DlERF10 protein.**

control group. A reduction was observed on 1 d, though followed by another rise on 3 d. The *DlERF10* expression in the low-temperature stress treatment on 3 d was 3.56 times the expression in the control group. A reduction occurred on 5 d, and the expression in the low-temperature stress treatment was 1.5 times the expression in the control group (Fig 4A). Under the drought stress using PEG6 00, the *DlERF10* expression increased from 6 h to 12 h, followed by a reduction from 1 d to 3 d. Another increase was observed on 5 d, and the expression increased most significantly on 5 d, reaching the peak. At this time point, the *DlERF10* expression in the drought stress treatment was 6.70 times the expression in the control group (Fig 4B). Under the NaCl-imposed salt stress, the relative *DlERF10* expression varied in a similar pattern as under the drought stress. The *DlERF10* expression peaked on 5 d, the value being 3.11 times the expression in the control group (Fig 4C). From the above it was speculated that *DlERF10* expression was induced by low temperature, drought, and salt stresses and *DlERF10* might be involved in the response of *D. lotus* to abiotic stresses.

Table 2. *Cis*-acting elements of *DlERF10* gene.

| Type of *cis*-acting element | Associated element | Number | Function of response |
|---|---|---|---|
| **Light response-related *cis*-element** | G-box | Three | Light-responsive element |
| | Box 4 | Three | Light-responsive element |
| | I-box | Two | Light-responsive element |
| | GATA-motif | One | Light-responsive element |
| | TCT-motif | Four | Light-responsive element |
| | AE-box | One | Light-responsive element |
| **Hormone response-related *cis*-element** | TGA-box | One | Growth hormone responsiveness |
| | P-box | One | Gibberellin responsiveness |
| | TCA-element | One | Salicylic acid responsiveness |
| | ABRE | One | Abscisic acid responsiveness |
| | TGACG-motif | Two | MeJA responsiveness |
| | CGTCA-motif | Two | MeJA responsiveness |
| **Growth-related *cis*-element** | CAT-box | One | Hyphal tissue expression-related |
| | circadian | One | Circadian regulatory |

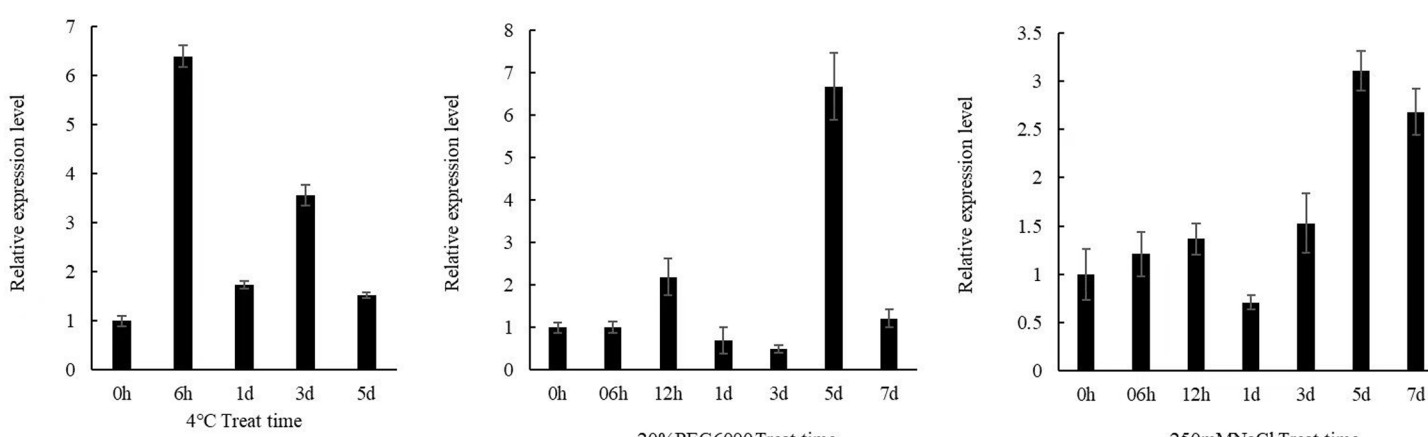

**Fig 4. Relative expression level of *DlERF10* gene under abiotic stress.**

## Expression vector construction and genetic transformation

The *DlERF10* gene was cloned from the leaves of *D. lotus* and used to construct the recombinant overexpression vector pBI121 (CaMV 35S promoter). The sequencing showed that the length of the nucleotide sequence of the cloned *DlERF10* gene was consistent with the search result. The coding sequence of the gene was also consistent with the reference sequence downloaded from NCBI, indicating that the expression vector was successfully constructed. Agrobacterium-mediated transformation was performed to transfer the gene into tobacco leaves (Fig 5). Two plants, namely, plant 11 and plant 35, were selected by screening on the kanamycin-containing culture medium (Fig 6). The DNA extracted from the transgenic tobacco plants was used as a template for PCR. The PCR-amplified bands appeared at similar

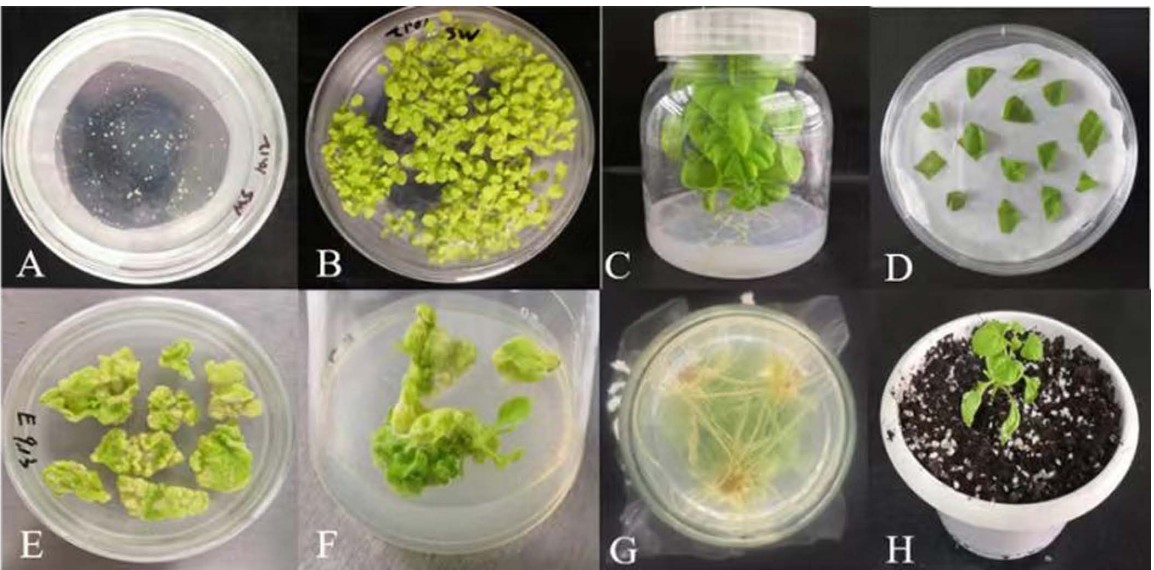

**Fig 5. Genetic transformation of tobacco with *DlERF10* overexpression vector.** (A) Tobacco seed seeding. (B-C) Sterile tobacco seedlings. (D) Co-culture. (E-F) Selective culture. (G) Rooting culture. (H) Transplanting.

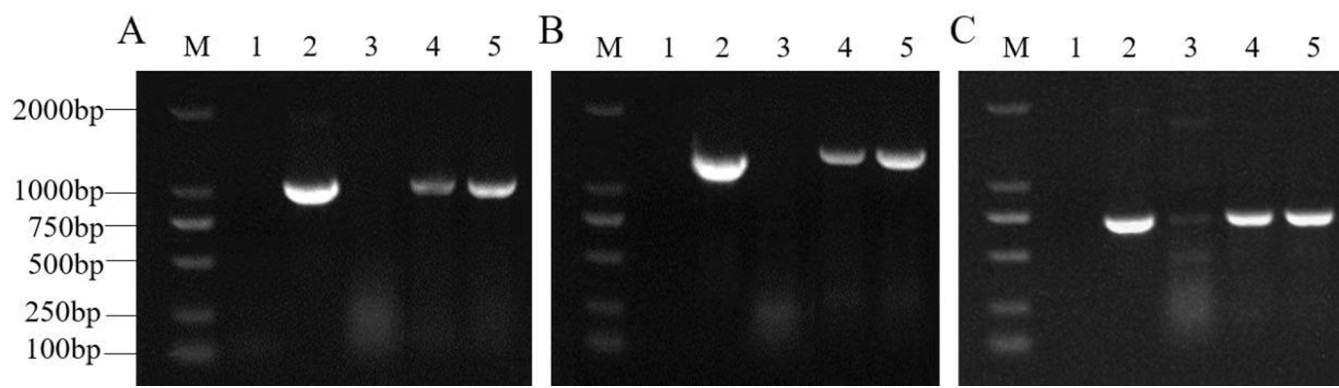

**Fig 6. PCR detection of transgenic tobacco plants.** (A) PCR validation of primers of the target gene. (B) PCR validation of 35S-F and downstream primer of the target gene. (C) PCR validation of the NPTII primers; M: DNA marker DL2000; 1: Negative control (water); 2: Positive control (plasmid); 3: Wild-type *N. benthamiana*; 4: Transgenic plant 11; 5: Transgenic plant 35.

positions as those in the positive control (plasmid). Therefore, the two plants were considered as positive (Fig 6).

## Phenotypic analysis of the transgenic tobacco plants

Potted tobacco seedlings in good growth and at the same age were chosen for phenotypic observation before the stress was imposed. As shown in Fig 7A, the leaves were dark green in the transgenic plants, while the leaves were light green in the wild-type plants. Plant heights were measured, and the transgenic plants were much shorter than the wild-type plants (Fig 8), indicating that *DlERF10* overexpression had an impact on plant growth.

At 6 h of low-temperature stress treatment (Fig 7B), leaves in the lower part of the wild-type and the transgenic plants were mildly wilted, but only with limited damage. On 1 d of low-temperature stress treatment (Fig 7C), except for the curled edges of two leaves in the top of the wild-type plants, all other leaves were wilted and drooping. The leaves in the lower part of the transgenic plant 11 were wilted and drooping and those in the upper part were only wilted at the edges but not drooping. In the transgenic plant 35, very few leaves in the lower part were wilted. On 3 d of low-temperature stress treatment (Fig 7D), the stalk of the

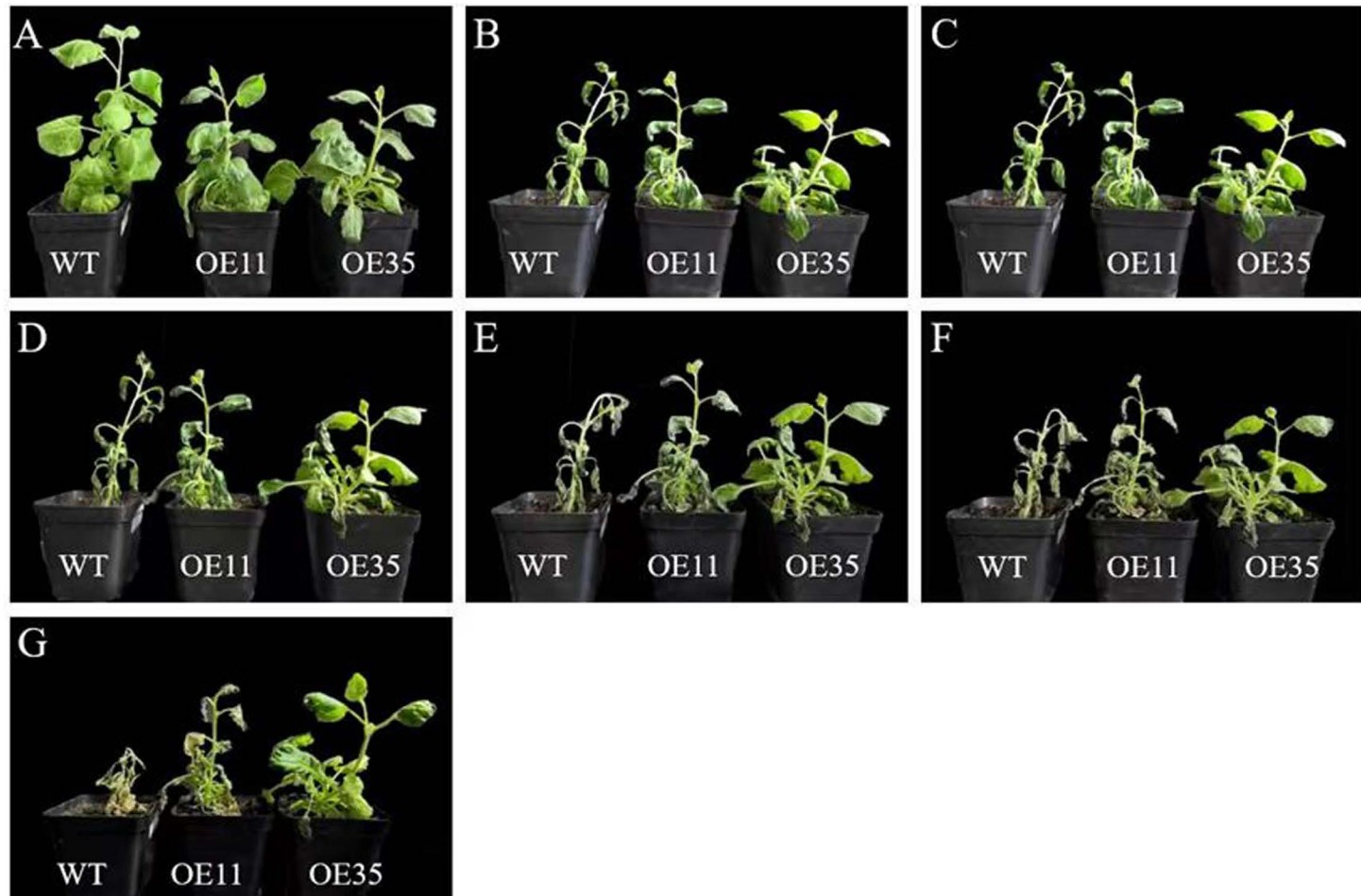

**Fig 7. Phenotypic changes of tobacco plants under low-temperature stress.** (A-F) Phenotype of tobacco plants treated at 0 h, 6 h, 1 d, 3 d, 5 d and 7 d under low-temperature stress. (G) Phenotype of tobacco plants after 2 days of recovery.

wild-type plants began to bend. In the transgenic plant 11, except for curled edges in two leaves in the top, all other leaves were wilted and drooping. A few leaves in the lower part of the transgenic plant 35 were wilted and drooping, while the leaves in the upper part remained normal. On 5 d of low-temperature stress treatment (Fig 7E), the apical buds of the wild-type plants began to wilt. The upper stalk segment was seriously curved into a right angle. Except for the apical bud, all other leaves began to wilt and droop in the transgenic plant 11. The leaves in the lower part of the transgenic plant 35 also began to wilt and droop, while those in the upper part were slightly wilted. On 7 d of low-temperature salt treatment (Fig 7F), the upper stalk of the wild-type plant was drooping. The apical bud in the transgenic plant 11 was slightly wilted, and the stalk began to curve. The leaves in the upper part of the transgenic plant 35 began to curl and dry out, while the apical bud was in good condition. After 2 d of recovery culture (Fig 7G), the death rate of wild-type plants was 30.8%. The death rate of the transgenic plant 11 was 23.08%, and the leaves of undead plants were severely dried up. The death rate of the transgenic plant 35 was 23.08%, and very few leaves of the undead plants were dried out, presenting fast growth during recovery. To sum up, the wild-type tobacco plants under low-temperature stress were severely wilted, drooping, and dried out, and their death rate was higher than that of the transgenic plants. The above results indicated that *DlERF10* overexpression enhanced the cold tolerance of the transgenic tobacco plants. The leaves of the transgenic plant 35 displayed even higher cold tolerance, and the plant suffered from less low-temperature-induced damage.

## Cold tolerance analysis of the transgenic tobacco plants

The MDA contents of the wild-type and transgenic tobacco plants remained low under normal conditions, without significant difference. As the duration of low-temperature stress was extended, the MDA contents of leaves in both wild-type and transgenic tobacco plants increased significantly. Meanwhile, the MDA content of the transgenic plants was significantly lower than that of the control group (Fig 9). On 3 d of low-temperature stress, the MDA content of leaves in the wild-type plant was 1.55 times the content in the transgenic plant 11 and 1.76 times the content in the transgenic plant 35. On 5 d of low-temperature

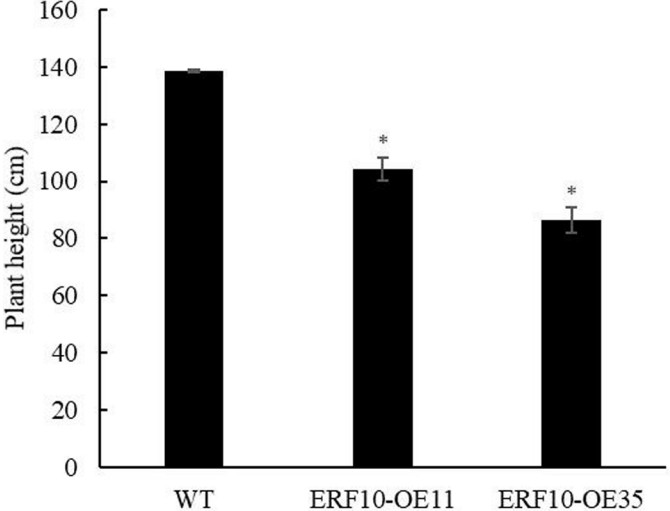

**Fig 8. Determination of plant height of tobacco.** * Means significant difference (P < 0.05).

stress, the MDA content of leaves in the wild-type plant was 1.62 and 1.92 times the content in the two transgenic plants, respectively. These results indicated that the cell membrane was less damaged in the transgenic plants.

Under normal conditions, the activities of SOD, POD, and CAT in the leaves of the wild-type and transgenic plants were not significantly different, and they were slightly higher in the transgenic plants. As the duration of low-temperature stress was extended, the activities of SOD, POD and CAT were significantly higher in the transgenic plants than in the wild-type plants. On 5 d of low-temperature stress, the SOD activity of both wild-type and transgenic plants peaked. In the wild-type plants, the SOD activity on 5 d was 1.71 times the activity before stress. The SOD activity of the two transgenic plants was 3.51 and 3.42 times the activity before stress, respectively, and 2.32 and 2.18 times the activity in the wild-type plant on 5 d (Fig 10A), respectively. On 3 d of low-temperature stress, the POD activity of the two transgenic plants was 1.40 and 1.53 times the activity in the wild-type plant, respectively (Fig 10B). The CAT activity of the two transgenic plants was 1.32 and 1.23 times the activity in the wild-type plant, respectively, on 3 d (Fig 10C). The above results suggested that *DlERF10* overexpression enhanced the antioxidant enzyme activities in the transgenic tobacco plants, thereby strengthening the reactive oxygen species scavenging capacity and hence the cold tolerance of the transgenic plants. The Pro contents of leaves were determined in the wild-type and transgenic tobacco plants (Fig 10D). Under low-temperature stress, the Pro content increased to varying degrees in all plants, though the Pro contents of the two transgenic plants were significantly higher than that of the wild-type plant. On 5 d of low-temperature stress, the Pro contents peaked in all plants. The Pro content in the two transgenic plants was 2.14 and 1.77 times the content before stress, respectively, and 1.34 and 1.19 times the content in the wild-type plant on 5 d, respectively. These results suggested that *DlERF10* overexpression promoted Pro accumulation in tobacco plants under low-temperature stress while maintaining the balance of external osmotic pressure in plant cells, thereby enhancing the cold tolerance of the transgenic plants.

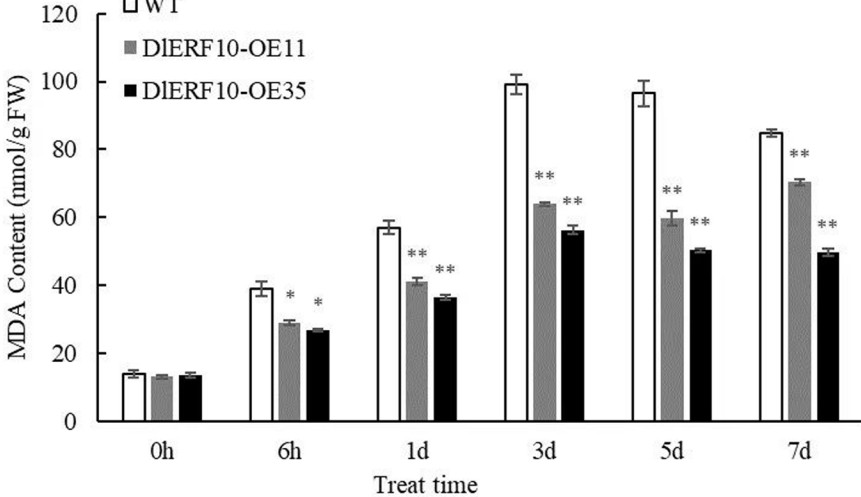

**Fig 9. Changes of MDA content in DlERF10 overexpressed tobacco and wild-type tobacco under low-temperature stress.** * means significant difference (P < 0.05), ** means extremely significant difference (P < 0.01). * and ** represent the same meaning unless otherwise specified.

## Discussion

AP2/ERF transcription factors are believed to play important regulatory roles in plant growth and stress responses. ERF proteins are part of the AP2/ERF family and are divided into the ERF and DREB subfamilies, each of which contains at least one AP2 DNA binding domain [80]. In the present study, an AP2/ERF family gene was cloned from *D. lotus*. The sequence analysis showed that the gene only contained one AP2 domain but no B3 domain. It was inferred that the protein encoded by the gene belonged to the ERF family. In all ERF subfamily proteins, the 14th and 19th amino acids in the AP2 domain are alanine and aspartic acid, respectively. These two amino acid residues play vital roles in specific DNA binding. In DREB subfamily proteins, the amino acids at the above two positions are valine and glutamic acid, respectively. The functions fulfilled by the ERF and DREB subfamily proteins are disparate due to their differences in the structural domain [81]. In this study, we found that the structural domain in the cloned gene sequence was typical of an ERF subfamily protein. Homologous cloning was performed to isolate the gene from *D. lotus*, using primers that were designed by alignment against the sequence of the *DkERF10* gene in *D. kaki* (GenBank accession No.: JX145122.1). Sequence analysis showed that the protein sequence encoded by this gene had the highest homology with the DkERF10 protein. Therefore, this gene was named *DlERF10*. The phylogenetic tree analysis indicated a close genetic relationship between the

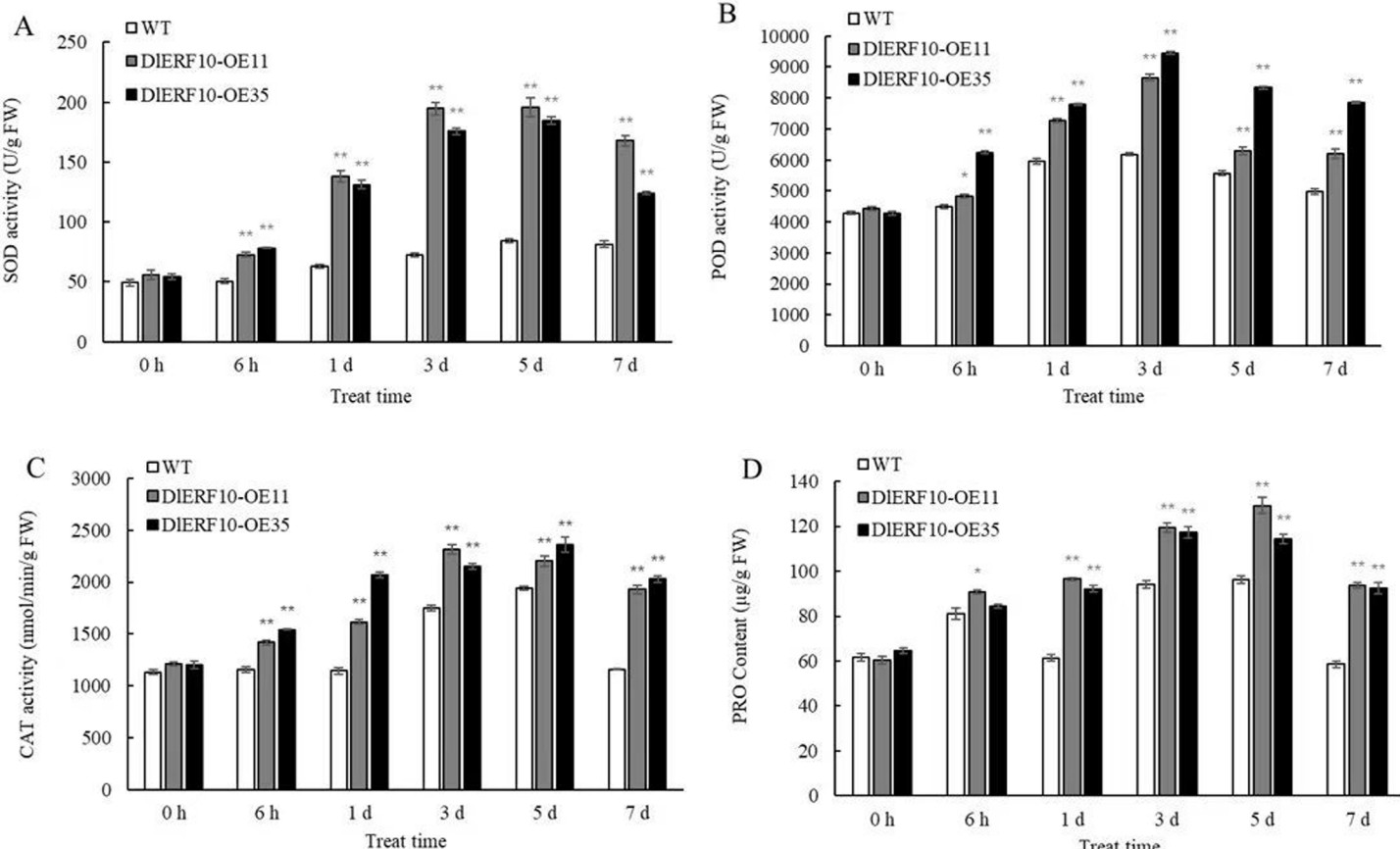

**Fig 10. Changes of SOD (A), POD (B), CAT (C) and PRO (D) of overexpressed *DlERF10* tobacco and wild-type tobacco under low-temperature stress.**

two species. It is generally believed that genes with high homology may share similar origins and perform similar biological functions.

Studies on the ERF transcription factors in *Diospyros* species have been generally focused on deastringency. For example, *DkERF1* and *DkERF6* can be only induced by $CO_2$ treatment, while *DkERF4* and *DkERF5* can be induced by both $CO_2$ and $C_2H_4$ treatments [82]. Among the hypoxia-responsive genes, the expression of three ethylene response factor genes (DkERF8/16/19) showed significant correlations with postdeastringency persimmon fruit softening [83]. Using RNA-seq and realtime PCR, twelve ethylene response factor genes (DkERF11-22) were isolated and characterized, to determine those responsive to high $CO_2$ treatment. Only two genes, DkERF19 and DkERF22, showed trans-activation effects on the promoters of deastringency-related genes pyruvate decarboxylase genes (DkPDC2 and DkPDC3) and the transcript levels of these genes was enhanced by hypoxia. Moreover, *DkERF19* and the previously isolated DkERF9 had additive effects on activating the DkPDC2 promoter [84]. According to another study, the ERF and NAC transcription factors were synergistically involved in cellulose and hemicellulose degradation during post-deastringency persimmon fruit softening [85]. Zhu [86] found that high $CO_2$/hypoxia-responsive transcription factors DkERF24 and DkWRKY1 interacted with each other, activating the DkPDC2 promoter. *DkERF19*, *DkERF23*, *DkERF24* and *DkERF25* were sensitive to hypoxia and might be also involved in hypoxia-driven deastringency and maturation of persimmon fruits. Besides, the transcription complex of *DkERF24* and *DkWRKY1* activated *DkPDC2*, which further affected the hypoxia response and deastringency of persimmon fruits. It has been found that the ERF subfamily proteins play vital roles in resisting abiotic stresses [87]. Wang [88] identified 57 low-temperature-responsive ZmERFs from the transcriptome of maize leaves constructed under 5 and 22°C, including 53 upregulated ZmERFs and 4 down-regulated ZmERFs. Real-time fluorescence quantitative PCR showed that the expression of *GRMZM2G434203* gene was induced under low temperature and drought stresses. The expression of the *GRMZM2G544539* and *GRMZM2G040664* genes was induced under low temperature, salt, and drought stresses. Jin [89] found that 'Jinpeng No. 1' tomato variety had a significant increase in the expression of the *SlERFb.2* gene under low-temperature stress at 4°C, which peaked after 4 h of low-temperature stress. *SlERFb.2* expression could be also induced by drought stress. Gao [47] performed RT-qPCR, which found that the expression of the *PpcERF5* gene was induced by low temperature. The peak expression was observed in the leaves of *Prunus pseudocerasus* after 4 h of low-temperature stress at 4°C. Expression of the *PpcERF5* gene promoted seed germination and precocious flowering of *Arabidopsis thaliana*, indicating the regulatory function of the gene in the dormancy release of flower buds of *Prunus pseudocerasus* [90]. In this study, real-time fluorescence quantitative PCR was performed to determine the expression of the *DlERF10* gene in the leaves of *D. lotus* under different abiotic stresses. The results showed that the expression first increased and then decreased under low temperature, drought and salt stresses. It can be said that the *DlERF10* gene from *D. lotus* was responsive to all three stresses. We have reason to suppose that the ERF transcription factors are involved in plant responses to various abiotic stresses, including low temperatures. We analysed the performance of transgenic tobacco under low-temperature stress and found that the transformed plants improved tolerance to low temperature. ERF transcription factors enhance plant cold tolerance by regulating downstream expression of cold tolerance-related genes. We analysed the pathways by which ERF transcription factors regulate genes downstream of low-temperature stress. One is through the regulation of key genes for plant hormone transduction in order to control hormone signalling pathways. PtrERF109 directly regulates *PtrPrxl* to scavenge ROS in plants for cold tolerance [91]. Secondly, cold tolerance is improved by regulating sugar metabolism pathway. *PtrBAM1* is an amylase synthesis gene in

citrus, and PtrCBF can regulate cold tolerance by regulating *PtrBAM1* expression and affecting soluble sugar levels [92]. Another AP2/ERF transcription factor encoding gene in citrus, PtrERF108 regulates cottonseed sugar content through direct regulation of *PtrRafS* (RafS), which regulates cold hardiness [93]. Third, collaboration with proteins to improve plant cold tolerance. EjERF39 can form a synergistic activation complex with EjMYB8 to enhance *Ej4CL1* gene expression, increase fruit lignification, and resist cold stress [63].

The MDA content is closely related to the degree of lipid peroxidation of cell membranes. MDA generation and accumulation can cause damage to plant cell membranes [94]. For this reason, MDA is a common physiological indicator in studies on plant aging and stress tolerance. Measuring MDA content offers a pathway to understanding lipid peroxidation of plant cell membranes, which makes it possible to assess stress tolerance indirectly. Redox reaction is the most important metabolic pathway in organisms, supplying energy for life sustenance and exerting a decisive impact on aging and death. Oxygen is essential to all life activities. However, some metabolic processes involving oxygen may result in the generation of cytotoxic by-products, such as oxygen radicals, or reactive oxygen species [95]. SOD is a major antioxidant enzyme, which transforms reactive oxygen species into hydrogen peroxide and water through disproportionation, thereby clearing away reactive oxygen species, protecting the cell membrane system against damage, and maintaining normal physiological functions of cell membranes [96]. POD is an oxidase found extensively in animals, plants, and microorganisms, and catalyzes a variety of oxidation reactions involving hydrogen peroxide [97]. As an enzyme scavenger, CAT is a common enzyme found in nearly all living organisms exposed to oxygen and catalyzes the decomposition of hydrogen peroxide to water and oxygen, protecting cells against hydrogen peroxide-induced damage [98]. SOD, POD, and CAT interact with each other and jointly participate in the removal of reactive oxygen species, protecting plant cells from the damage caused by reactive oxygen species. Pro is an osmoregulatory substance that maintains osmotic balance across membranes within the body and structural integrity of membranes. Pro can also activate SOD, POD, and CAT to clear away reactive oxygen species [99]. The amount of Pro accumulating in plants is an indicator of stress tolerance, to a greater or lesser degree. The Pro content is also a common measure of stress tolerance in plants [100]. In the present study, the MDA content of leaves of the transgenic tobacco plants under low-temperature stress was lower than that of the wild-type plant. Meanwhile, the SOD, POD, CAT, and Pro contents were higher in the former than in the latter. This indicated that compared with the wild-type tobacco plants, the transgenic plants suffered from less cell membrane damage under low-temperature stress. Higher activities of antioxidant enzymes usually mean a greater reactive oxygen species scavenging capacity and less disturbance caused by low temperatures to cellular osmolality and cell membranes. As a result, the transgenic tobacco plants displayed greater cold tolerance, which agreed with previous findings in grapes [65, 66], bananas [101], and birch [102].

## Conclusion

In this study, the *DlERF10* gene was cloned from the leaves of *Diospyros lotus* L., with a full length of 1104 bp and encoding a sequence consisting of 367 amino acids. The coded protein contained one transmembrane structural domain and a typical conserved AP2 domain and belonged to the ERF subfamily. Under the low temperature, drought and salt stresses, the *DlERF10*gene was upregulated, indicating that the expression of the *DlERF10* gene was induced by different stresses and the gene might act as a multi-functional transcription regulation factor. The contents of MDA and Pro and the activity of SOD, CAT, POD in the transgenic tobacco plants were determined under low-temperature stress, and the results

confirmed the superiority of the transgenic plants in cold tolerance. Our research lays the foundation for revealing the regulatory function of the *DlERF10* gene under abiotic stresses.

## Supporting information

**S1 Fig.** "S1_raw_images.pdf" is the raw images.
(PDF)

## Author contributions

**Conceptualization:** Ruijin Zhou, Huiling Hu.

**Data curation:** Shuda Li, Xiaona Zhang, Yingying Wang.

**Formal analysis:** Ruijin Zhou, Shuda Li, Xiaona Zhang, Yingying Wang.

**Investigation:** Shuda Li, Xiaona Zhang, Yingying Wang.

**Writing – original draft:** Ruijin Zhou.

**Writing – review & editing:** Ruijin Zhou, Huiling Hu.

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
