## [Decision Letter · Decision Letter 0]

21 Aug 2024

PONE-D-24-28978Cloning of the DlERF10 gene from Diospyros lotus L. and cold resistance analysis of the DlERF10 gene in transgenic tobacco plantsPLOS ONE

Dear Dr. Zhou,

Thank you for submitting your manuscript to PLOS ONE. After careful consideration, we feel that it has merit but does not fully meet PLOS ONE’s publication criteria as it currently stands. Therefore, we invite you to submit a revised version of the manuscript that addresses the points raised during the review process.

We look forward to receiving your revised manuscript.

Kind regards,

Diaa Abd El-Moneim

Academic Editor

PLOS ONE

4. We note that Figure(s) 12 and 14 in your submission contain copyrighted images. All PLOS content is published under the Creative Commons Attribution License (CC BY 4.0), which means that the manuscript, images, and Supporting Information files will be freely available online, and any third party is permitted to access, download, copy, distribute, and use these materials in any way, even commercially, with proper attribution. For more information, see our copyright guidelines: http://journals.plos.org/plosone/s/licenses-and-copyright.

a. You may seek permission from the original copyright holder of Figure(s) 12 and 14 to publish the content specifically under the CC BY 4.0 license. 

5. Please remove your figures from within your manuscript file, leaving only the individual TIFF/EPS image files, uploaded separately. These will be automatically included in the reviewers’ PDF.

Reviewers' comments:

Reviewer's Responses to Questions

**Comments to the Author**

1. Is the manuscript technically sound, and do the data support the conclusions?

Reviewer #1: Yes

Reviewer #2: Yes

2. Has the statistical analysis been performed appropriately and rigorously? 

Reviewer #1: Yes

Reviewer #2: Yes

3. Have the authors made all data underlying the findings in their manuscript fully available?

Reviewer #1: Yes

Reviewer #2: Yes

4. Is the manuscript presented in an intelligible fashion and written in standard English?

Reviewer #1: No

Reviewer #2: Yes

5. Review Comments to the Author

Reviewer #1: Cloning of the DlERF10 gene from Diospyros lotus L. and cold resistance analysis of the DlERF10 gene in transgenic tobacco plants

Line 95: there is no appropriate reference for text: Xu et al. 2021

Line 237-239: there is no figure legend for figure 1

Line 241-242: what is second conserve DLERF10 protein (the figure 2 legend is ambiguous)

Line 314: what is the word and its number: proteins[45].

Line 351: figure 11 - there are no determined what figure is A, B or C

Line 488: please write correct the word: D. lous.

Line 497: has been demonstrated in previous studies.......what is previous study?

Line 536: The Gao YD et al. (2020) is no searchable

Line 562: The Wei et al (2020) is no searchable

This reference not used in the text:

Ohme-Takagi M, Shinshi H. Ethylene-inducible DNA binding proteins that interact764 with an ethylene-responsive element. The Plant Cell, 1995, 7(2): 173-182. DOI:765 10.1105/tpc.7.2.173.766

The discussion odf manuscript is very tall and boring for the readership

Sincerely yours

Bahram Kazemi PhD, Professor at Shahid Beheshti University of Medical Sciences

Reviewer #2: In this manuscript (PONE-D-24-28978) by Li, et al., nicely describes the Cloning of the DlERF10 gene from Diospyros lotus L. and cold resistance analysis of the DlERF10 gene in transgenic tobacco plants. Overall, I think this manuscript should be accepted after minor revision. but I only have a few minor concerns:

1.There are too many Figures in this manuscript, some figures should be placed together, thus the number of the figures could be set no more that 10.

2.There are many other transcription factors involved in abiotic stress responses, such as cold, salt and drought stress. So in this paper, more studies of WRKY, MYB, bHLH, or NAC in abiotic stress should be introduced in this MS.

3.More references of WRKY transcription factors in the regulation of biological or abiotic stress should be included in the revised manuscript, such as MbCBF2 (International Journal of Molecular Sciences. 2022, 23, 9827), MbERF12 (In Vitro Cellular & Developmental Biology - Plant. 2021, 57: 760–770) and MbWRKY4 (International  Journal of  Agriculture & Biology. 2018, 20: 2045–2052) from Malus baccata; MxWRKY55 (In Vitro Cellular & Developmental Biology - Plant. 2020, 56: 600–609), and MxWRKY53 (In Vitro Cellular & Developmental Biology - Plant. 2022, 58: 266–278) from Malus xiaojinensis; Furthermore, FvICE1 (Plant Physiology and Biochemistry. 2023, 196:270–280) from Fragaria vesca.

4.There should have bars in figure 12, and 14.

5.The Chinese characters should not appear in the figures.

6.Subcellular localization results should be added to the submission.

7.The expression levels of DlERF10 were studied under cold, drought and salt, why only the cold tolerance was researched?

8.The word “resistance” should changed into “ tolerance”.

9.If the figures were combined with more that 1 part, every part should be marked with A, B, C, and so on.

10.The quality of the written English should be improved. In this regard, you may request the assistance from a company specialized in scientific editing.

6. PLOS authors have the option to publish the peer review history of their article (what does this mean? ). If published, this will include your full peer review and any attached files.

**Do you want your identity to be public for this peer review?** For information about this choice, including consent withdrawal, please see our Privacy Policy .

Reviewer #1: No

Reviewer #2: No

---

## [Author Response · Author response to Decision Letter 1]

10 Oct 2024

Dear editor,

RE: Cloning of the DlERF10 gene from Diospyros lotus L. and cold tolerance analysis of the DlERF10 gene in transgenic tobacco plants (Manuscript ID: PONE-D-24-28978)

Thank you for your email of 21st August 2024 and for providing us an opportunity to submit a revised manuscript to POLS ONE. We are also grateful to the Editor and two reviewers for their time and useful comments on the manuscript. Based on these comments, we have revised the manuscript. The revisions and responses to the reviewers’ concerns are outlined in details below. We hope these revisions are sufficient and look forwards to hearing from you for a positive outcome in the near future.

We would like to thank the reviewers again for taking the time to review our manuscript!

Yours sincerely

Rui-Jin Zhou

Reviewer#1:

1. Line 95: there is no appropriate reference for text: Xu et al. 2021

Response. Thank you for pointing this out. We have added the corresponding references. Line 98 and Lines 699-702.

The ZmERF1 gene in maize is involved in regulating hypoxia tolerance under waterlogging stress[61].

61. Xu Y, Lu JH, Zhang JD, Liu DK, Wang Y, Niu QD, et al. Transcriptome revealed the molecular mechanism of Glycyrrhiza inflata root to maintain growth and development, absorb and distribute ions under salt stress. BMC Plant Biology. 2021; 21: 599. doi: 10.1186/s12870-021-03342-6

2. Line 237-239: there is no figure legend for figure 1

Response. Thank you for pointing this out. We have added the figure legend for figure 1. Line 221. Fig 1. Result of PCR product of the purpose gene

3. Line 241-242: what is second conserve DLERF10 protein (the figure 2 legend is ambiguous)

Response. Thank you for pointing this out. We have revised the text and deleted figure 2.

4. Line 314: what is the word and its number: proteins[45].

Response. Thank you for pointing this out. We have added the corresponding references. Line 237 and lines 759-761.

79. Ohme-Takagi M, Shinshi H. Ethylene-inducible DNA binding proteins that interact with an ethylene-responsive element. The Plant Cell. 1995; 7(2): 173-182. doi: 10.1105/tpc.7.2.173

5. Line 351: figure 11 - there are no determined what figure is A, B or C.

Response. Thank you for the suggestion. We have added A, B and C in the figure.

Line 276.

6. Line 488: please write correct the word: D. lous

Response. Thank you for pointing this out. We corrected the spelling mistakes.

Line 377.

In the present study, an AP2/ERF family gene was cloned from D. lotus.

7. Line 497: has been demonstrated in previous studies.......what is previous study?

Response. Thank you for the suggestion. The previous expression was not very accurate, and we modified the statement.

Lines 383-384.

In this study, we found that the structural domain in the cloned gene sequence was typical of an ERF subfamily protein.

8. Line 536: The Gao YD et al. (2020) is no searchable.

Response. Thank you for pointing this out. We have added the corresponding references. Lines 416-417 and lines 645-648.

47. Gao Y, Han D, Jia W, Ma X, Yang Y, Xu Z. Molecular characterization and systematic analysis of NtAP2/ERF in tobacco and functional determination of NtRAV-4 under drought stress. Plant Physiology and Biochemistry. 2020; 156: 420-435. doi: 10.1016/j.plaphy.2020.09.027

9. Line 562: The Wei et al (2020) is no searchable.

Response. Thank you for pointing this out. We have added the corresponding references. Lines 819-821.

96. Wei J, Xu C, Li K, He H, Xu Q. Progress on superoxide dismutase and plant stress resistance. Plant Physiology Journal. 2020; 56(12): 2571-2584. doi: 10.13592/j. cnki.ppj.2020.0311

10. This reference not used in the text:

Ohme-Takagi M, Shinshi H. Ethylene-inducible DNA binding proteins that interact764 with an ethylene-responsive element. The Plant Cell, 1995, 7(2): 173-182. DOI:765 10.1105/tpc.7.2.173.766

Response. Thank you for pointing this out. We have marked the references in the original text. Lines 236-237.

which are typical of the ERF family proteins[79].

11. The discussion odf manuscript is very tall and boring for the readership.

Response. Thank you for the suggestion. We have revised the text to address your concerns and hope that it is now clearer. Please see page 15 of the revised manuscript, lines 374-395, page 16, lines 396-438, and page 17, lines 439-467.

Reviewer #2:

1.There are too many Figures in this manuscript, some figures should be placed together, thus the number of the figures could be set no more that 10.

Response. Thank you for the suggestion. We have revised the text to address your concerns and hope that it is now clearer. Please see page 8 of the revised manuscript, lines 223-248, and page 9, lines 249-251. And we have adjusted the figures in the paper to keep only 10.

2.There are many other transcription factors involved in abiotic stress responses, such as cold, salt and drought stress. So in this paper, more studies of WRKY, MYB, bHLH, or NAC in abiotic stress should be introduced in this MS.

Response. Thank you for the suggestion. In the manuscript we added the related research progress of WRKY, NAC and other transcription factors. However, due to the space requirement, only a brief introduction of the research progress was made, and only the ERF transcription factors in this study were described in detail. Please see page 2 of the revised manuscript, lines 32-34, 38-49, and page 3, lines 50-70.

3.More references of WRKY transcription factors in the regulation of biological or abiotic stress should be included in the revised manuscript, such as MbCBF2 (International Journal of Molecular Sciences. 2022, 23, 9827), MbERF12 (In Vitro Cellular & Developmental Biology - Plant. 2021, 57: 760–770) and MbWRKY4 (International Journal of Agriculture & Biology. 2018, 20: 2045–2052) from Malus baccata; MxWRKY55 (In Vitro Cellular & Developmental Biology - Plant. 2020, 56: 600–609), and MxWRKY53 (In Vitro Cellular & Developmental Biology - Plant. 2022, 58: 266–278) from Malus xiaojinensis; Furthermore, FvICE1 (Plant Physiology and Biochemistry. 2023, 196:270–280) from Fragaria vesca.

Response. Thank you for the suggestion. We added more references of WRKY transcription factors in the manuscript. Please see page 2 of the revised manuscript, lines 44-45, and page 19, lines 515-526.

4.There should have bars in figure 12, and 14.

Response. Thank you for the suggestion. At your suggestion, we have added bars in figure 5(12) and 7(14). Line 289 and line 304.

5.The Chinese characters should not appear in the figures.

Response. I'm sorry. It was an oversight. We have made changes. Line 276(Fig.4), line 309(Fig. 8), line 345(Fig. 9) and line 370(Fig.10).

6.Subcellular localization results should be added to the submission.

Response. Thank you for the suggestion. We reorganized the bioinformatics analysis section, and removed some figures. Subcellular localization results added to the submission. Lines 225-227.

It is a hydrophilic protein without a signal peptide structural domain, with a subcellular localization predicted to be in the nucleus, and a secondary and tertiary structure consisting mainly of an irregularly convoluted and α-helix.

7.The expression levels of DlERF10 were studied under cold, drought and salt, why only the cold tolerance was researched?

Response. Thank you for the suggestion. The expression levels of DlERF10 were studied under cold, drought and salt. In this paper, we only focused on describing cold tolerance, and we also studied other stresses, but the data were not presented in this paper.

8.The word “resistance” should changed into “ tolerance”.

Response. Thank you for the suggestion. At your suggestion, we have changed “resistance” into “tolerance”.

9.If the figures were combined with more that 1 part, every part should be marked with A, B, C, and so on.

Response. Thank you for the suggestion. We have marked with A, B, C, and so on in the figures. Line 276 and 370.

10.The quality of the written English should be improved. In this regard, you may request the assistance from a company specialized in scientific editing.

Response. Thank you for the suggestion. We have corrected typos and spelling errors in the manuscript. And we requested the assistance from a company specialized in scientific editing. Proof of editing is provided in the appendix.

---

## [Decision Letter · Decision Letter 1]

6 Nov 2024

Cloning of the DlERF10 gene from Diospyros lotus L. and cold resistance analysis of the DlERF10 gene in transgenic tobacco plants

PONE-D-24-28978R1

Dear Dr. Zhou,

We’re pleased to inform you that your manuscript has been judged scientifically suitable for publication and will be formally accepted for publication once it meets all outstanding technical requirements.

Kind regards,

Diaa Abd El-Moneim

Academic Editor

PLOS ONE

Additional Editor Comments (optional):

Reviewers' comments:

Reviewer's Responses to Questions

**Comments to the Author**

1. If the authors have adequately addressed your comments raised in a previous round of review and you feel that this manuscript is now acceptable for publication, you may indicate that here to bypass the “Comments to the Author” section, enter your conflict of interest statement in the “Confidential to Editor” section, and submit your "Accept" recommendation.

Reviewer #2: All comments have been addressed

2. Is the manuscript technically sound, and do the data support the conclusions?

Reviewer #2: Yes

3. Has the statistical analysis been performed appropriately and rigorously? 

Reviewer #2: Yes

4. Have the authors made all data underlying the findings in their manuscript fully available?

Reviewer #2: Yes

5. Is the manuscript presented in an intelligible fashion and written in standard English?

Reviewer #2: Yes

6. Review Comments to the Author

Reviewer #2: In the revised manuccript, All my concerns have been resolved. So I think this manuscript can be accepted now.

7. PLOS authors have the option to publish the peer review history of their article (what does this mean? ). If published, this will include your full peer review and any attached files.

**Do you want your identity to be public for this peer review?** For information about this choice, including consent withdrawal, please see our Privacy Policy .

Reviewer #2: **Yes: ** Deguo HAN

---

## [Editor Report · Acceptance letter]

PONE-D-24-28978R1

PLOS ONE

Dear Dr. Zhou,

I'm pleased to inform you that your manuscript has been deemed suitable for publication in PLOS ONE. Congratulations! Your manuscript is now being handed over to our production team.

Kind regards,

on behalf of

Dr. Diaa Abd El-Moneim

Academic Editor

PLOS ONE